# Identification and Functional Analysis of Tomato CIPK Gene Family

**DOI:** 10.3390/ijms21010110

**Published:** 2019-12-23

**Authors:** Yao Zhang, Xi’nan Zhou, Siyuan Liu, Anzhou Yu, Chuanming Yang, Xiuling Chen, Jiayin Liu, Aoxue Wang

**Affiliations:** 1College of Life Sciences, Northeast Agricultural University, Harbin 150030, China; zy13263696020@163.com (Y.Z.); Xinan15245116526@163.com (X.Z.); 18245699439@163.com (A.Y.); 2College of Plant Protection, China Agricultural University, Beijing 100000, China; liusiyuan6992@126.com; 3College of Arts and Sciences, Northeast Agricultural University, Harbin 150030, China; yangchuanming1980@163.com; 4College of Horticulture and Landscape Architecture, Northeast Agricultural University, Harbin 150030, China; chenx@neau.edu.cn; 5College of Sciences, Northeast Agricultural University, Harbin 150030, China; 13040216@163.com

**Keywords:** CIPK, tomato, abiotic stress, SlCIPK1 and SlCIPK8

## Abstract

The calcineurin B-like interacting protein kinase (CIPK) protein family is a critical protein family in plant signaling pathways mediated by Ca^2+^, playing a pivotal role in plant stress response and growth. However, to the best of our knowledge, no study of the tomato *CIPK* gene family in response to abiotic stress has been reported. In this study, 22 members of the tomato *CIPK* gene family were successfully identified by using a combination of bioinformatics techniques and molecular analyses. The expression level of each member of tomato *CIPK* gene family under abiotic stress (low temperature, high salt, drought treatment) was determined by qRT-PCR. Results indicated that tomato CIPK demonstrated different degrees of responding to various abiotic stresses, and changes in *SlCIPK1* and *SlCIPK8* expression level were relatively apparent. The results of qRT-PCR showed that expression levels of *SlCIPK1* increased significantly in early stages of cold stress, and the expression level of *SlCIPK8* increased significantly during the three treatments at different time points, implicating *Solanum lycopersicum* CIPK1(SlCIPK1) and *Solanum lycopersicum* CIPK8 (SlCIPK8) involvement in abiotic stress response. *SlCIPK1* and *SlCIPK8* were silenced using Virus-induced gene silencing (VIGS), and physiological indexes were detected by low temperature, drought, and high salt treatment. The results showed that plants silenced by *SlCIPK1* and *SlCIPK8* at the later stage of cold stress were significantly less resistant to cold than wild-type plants. *SlCIPK1* and *SlCIPK8* silenced plants had poor drought resistance, indicating a relationship between SlCIPK1 and SlCIPK8 with response to low temperature and drought resistance. This is the first study to uncover the nucleotide sequence for tomato CIPK family members and systematically study the changes of tomato CIPK family members under abiotic stress. Here, we investigate the CIPK family’s response under abiotic stress providing understanding into the signal transduction pathway. This study provides a theoretical basis for elucidating the function of tomato *CIPK* at low temperature and its molecular mechanism of regulating low temperatures.

## 1. Introduction

Plants are subject to various kinds of abiotic stress during their growth and development, such as high salt, drought, low temperature, and low potassium. However, many plants cannot actively avoid like many animals, so mechanisms have evolved to send emergency signals to adapt to stresses within their growing environment [1]. As a secondary messenger, Ca^2+^ broadly participates in the signaling during plant growth and development and the regulation of responding to environmental stimuli [2,3]. When plants suffer external stimulus, the concentration of Ca^2+^ in the cell fluctuates temporarily. The receptors on calcium ions receive this change and transmit this calcium signal downstream, thus causing a series of physiological and biochemical changes in the plant [4].

The receptors on calcium ions mainly include calmodulin protein (CaM), calmodulin like protein (CML), calmodulin-like B protein (CBL), calcium-dependent protein kinase (CDPK), and calmodulin-dependent protein kinase (CaMK) [5]. Among them, CBL is a particular kind of protein. Independent genetic analysis of Salt Overly Sensitive (SOS) mutants identified a gene known as SOS3 [6], which was placed in the CBL family and referred to as CBL4. CBL needs to specifically interact with a class of protein kinases (CBL-interacting protein kinase, CIPK) to activate downstream targets so as to regulate the physiological functions of plants [7,8,9,10]. With a yeast two-hybrid (Y2H) assay, Shi et al. (1999) identified a family of protein kinases as interactors of CBLs [11]. CBL interacting kinases are called CIPKs. According to the relationship between structural features and evolution, these kinases belong to the third class of SNF1-related protein kinases, group 3 (SnRK3) kinase.

The structure of the CBL proteins is conserved. All CBLs contain E-helix, F-helix, and Ca^2+^ binding loops. Phylogenetically, canonical EF hands constitute one of the most conserved structural elements and are responsible for high affinity Ca^2+^ binding [12,13,14]. Generally, CBL proteins contain 4 EF hands like calmodulin proteins (CaMs) [15], and so on. The analysis on structure of CIPK shows that CIPK includes the kinase domain at the N-terminal and the regulatory domain at the C-terminal, which contains a conserved NAF domain composed of 24 amino acids that is a self-inhibiting domain and contains a PPI domain capable of interacting with Type 2C Protein Phosphatases (PP2C) [16] to mediate the specific binding with CBL [17]. In adverse stresses conditions, Ca^2+^ signature binds to EF hands domains of the CBL proteins. Consequently, the CBL proteins bind the NAF/FISL domain of C-terminal in the CIPK to stimulate the kinase [18]. On the other hand, N-terminal of the CBL proteins directs the CBL-CIPK system to an exact cellular target region ensuing in the stimulated CIPK phosphorylating the proper target proteins [19].

Previous studies revealed that CBL can interact with one or more specific CIPKs, and also each CIPK interacts with one or more CBLs. CBLs interact with CIPKs forming a complex CBL-CIPK network in plant signal transduction [8,9]. The earliest study on the function of CBL-CIPK in plants stemmed from research into the SOS pathway. The interaction of CIPK24/SOS2 and CBL4/SOS3 was identified by using genetic screening and had a role in salt tolerance in *Arabidopsis* [20]. Studies have established that CBL4/SOS3–CIPK24/SOS2 might directly regulate SOS1 which is a putative Na^+^/H^+^ antiporter act in the downstream of SOS pathway [21], thereby enhancing the salt detoxification process. Studies on *Arabidopsis thaliana* found that AtCBL1-AtCIPK24 and AtCBL10-AtCIPK24 also participate in the regulation of high salt stress [22,23,24]. However, unlike CBL4/SOS3, which functions primarily in the roots [25], CBL10 is expressed and functions in the shoots and leaves [23,26], while CBL1 functions at the whole plant level. Compared with wild-type, the *Arabidopsis cipk6* mutant is more sensitive to salt stress, indicating that CIPK6 is also involved in the SOS pathway [27]. The cipk21 mutant of *Arabidopsis* displayed the phenotype of hypersensitivity to salt, and CIPK21 could interact with CBL2/3 to jointly regulate salt stress [28].

Currently, complete pathways of SOS signaling have been identified in both Rice and Poplar [29,30]. OsCBL4 is homologous with AtCBL4/SOS3 and is functionally complementary to the *sos3* mutant of *Arabidopsis*. In addition, AtCBL4/SOS3 is found to phosphorylate OsSOS1 protein [24]. Overexpression of OsCIPK15 can enhance the resistance of rice to salt stress [13]. ZmCIPK21 can be combined with ZmCBL2/ZmCBL3, and overexpression of ZmCIPK21 in *Arabidopsis* can enhance its salt resistance [31].

The CBL-CIPK signal transduction pathway plays a crucial role in cold stress. Studies on *Arabidopsis* indicate that AtCBL9-AtCIPK3 and AtCBL1-AtCIPK7 play important roles in low-temperature signaling pathways [32,33]. The expression of OsCIPK3 in rice increased significantly under cold induction, which can enhance the cold resistance of rice [34]. Sun et al. (2015) found that root and leaf responses to low-temperature stress were induced by *Triticum aestivum L. CBL4* (*TaCBL4), Triticum aestivum L. CBL9 (TaCBL9), Triticum aestivum L. CIPK7 (TaCIPK7), Triticum aestivum L. CIPK15(TaCIPK15*) and *Triticum aestivum L. CIPK24(TaCIPK24)* [35]. Furthermore, CIPK14 in wheat was induced by low temperature, and the heterotopic expression of wheat TaCIPK14 in tobacco could enhance its resistance to cold stress [36].

Under drought stress, AtCBL1 and AtCBL9 can interact with AtCIPK1 to maintain the osmotic balance of *Arabidopsis* cells [37]. Overexpression of AtCBL5 can improve its resistance to osmosis and drought stress [38], and so do OsCIPK12 and OsCIPK23 [34,39]. After drought treatment, the expression of ZmCIPK8 in roots and leaves changed significantly, and overexpression of ZmCIPK8 in tobacco improves the tolerance under drought stress, and there is an interaction in ZmCIPK8 and ZmCBL1, ZmCBL4, and ZmCBL9 [40].

Except for abiotic stress, CBL-CIPK is also involved in nutrient absorption and transport. This was demonstrated that CBL1 and CBL9 can both target CIPK23, which functions in the regulation of potassium (K) uptake and stomatal movements [17,41,42,43]. Chen et al reported that the CBL-CIPK system of cabbage type rape responds to low phosphorus stress. At low phosphorus levels, the expression of BnCBL1 and BnCIPK6 are up-regulated. Moreover, overexpressed BnCBL1 or BnCIPK6 in *Arabidopsis* showed more vigorous growth of lateral roots and more biomass accumulation than the wild-type [44]. Reports show the roles of *Arabidopsis* CIPK3 CIPK9, CIPK23 and CIPK26, which regulate the balance of magnesium ions in downstream CBL2/3 plants [45]. Researches have reported that AtCIPK26 can interact with *Arabidopsis thaliana* Respiratory Burst Oxidase Homolog F (AtRbohF) to inhibit the production of ROS, AtCIPK11 can directly interact with the proton pump on the cell membrane to inhibit the activity of H^+^-ATP enzyme H(+)-ATPASE 2 (AHA2), and *Atcipk11* deletion mutants increased AHA2 activity and plant tolerance to high pH stress, while CIPK1 can interact with CBL1 and CBL9, respectively, to influence plant response to ABA [46,47,48,49]. The functional specificity defined by different CBL-CIPK complexes demonstrates the diversity of CBL-CIPK network functions.

Tomato is an important economic crop, recognized for its fruit, with good nutritional value, widely planted around the world. Tomato is also a model plant for evolutionary, genetics, and stress tolerance studies. CBL-CIPK signaling pathway has been reported in tomato in order to cope with stresses. Studies indicated that under high salt stress, overexpressing SlSOS2 showed stronger tolerance than the wild-type [50]. SlSOS1 not only participates in the regulation of ion homeostasis under high salt stress, but also plays a vital role in the distribution of Na^+^ between plant organs [51]. Tomato CBL10 and CIPK6 mediate the signaling network regulated by calcium ions, and the interaction with Respiratory Burst Oxidase Homolog B (RbohB) plays a vital role in the tomato defense response [52].

CIPK, as a significant protein family in the calcium ion signal transduction pathway, has not been reported in tomatoes. In this study, to determine the sequence information of members of the tomato CIPK gene family, the amino acid sequence of CIPK gene family of *Arabidopsis* was inferred from the tomato whole genome, and the sequence information of members of tomato CIPK gene family were identified and analyzed using various bioinformatic methods. The expression of changes of each member under abiotic stress were studied using qRT-PCR. Virus-induced gene silencing technology verified the function of SlCIPK1 and SlCIPK8 in abiotic stress (low temperature, high salt, drought). These studies provide new ideas for the further exploration of calcium signaling pathways in tomato under abiotic stress.

## 2. Results

### 2.1. Identification and Phylogenetic Analysis of Tomato CIPK Genes

In order to identify the members of tomato CIPK family, we conducted a localized tBLASTn comparison between the nucleotide sequences of the existing tomato annotation genes and the CIPK amino acid sequences of *Arabidopsis thaliana* (E-value < 1 × 10^−10^, identity > 50%), removing duplicates to obtain tomato candidate CIPK. We used the Prosite (http://prosite.expasy.org/) website to identify candidate CIPK proteins domain, screening CIPKs with NAF and PPI domain structure, and eventually received 22 members of CIPK family (Table 1). All the 22 *SlCIPK* genes are distributed across all chromosomes except for chromosome X, and chromosome V has the most members (Figure 1). As for the SlCIPK proteins, the molecular weight of the predicted proteins ranged from 47.6 to 54.6 kDa (Table 1). We also found that the amino acid sequence identity of different SlCIPKs ranged from 53% to 84%. The deduced amino acid sequences of 22 *SlCIPK* genes demonstrated great conservation in size, which ranged from 419 to 478. As in CIPKs from *Arabidopsis*. All SlCIPKs consisted of a conserved N-terminal kinase domain, followed by a variable junction domain and a C-terminal regulatory domain. The divergent regulatory domain consists of a conserved NAF or FISL motif required for mediating CBL interaction and other motif, called PPI motif, mediating the interaction with type 2C protein phosphates. Comparison of the 24-amino acid NAF domains of 22 potential SlCIPKs demonstrated that residues are conserved, suggesting that these residues may play an important role in interacting with CBLs. According to the phylogenetic analysis of tomato, *Arabidopsis*, poplar, wheat, rice, and canola, CIPKs can be divided into A, B, C, D, E, F, G, and H subtribes.

### 2.2. Analysis of Tomato CIPK Gene Conserved Motif

Motif analysis further supported the phylogenetic relationship and classification of tomato CIPKs. In Figure 2, fifteen conserved motifs of tomato CIPKs were captured by motif analysis using MEME software and subsequent annotation with InterPro (Figure 2). Among them, SlCIPK1, SlCIPK2, SlCIPK7, SlCIPK8, SlCIPK19 have the same conservative motif and order. SlCIPK4, SlCIPK13, SlCIPK15, SlCIPK16, SlCIPK21 have the same conservative motif and order.

### 2.3. Expression Analysis and GO Analysis of Tomato CIPKs Gene

In order to study the tomato *CIPK* gene more systematically, Gene Ontology (GO) analysis was carried out to predict the role of the gene family in plant growth and development (Figure 3). It was found that the gene family was mainly localized to the plasma membrane and nucleus, and a small amount was also distributed in the vacuole membrane, plastid body, and cytoplasm. Regarding molecular function, about 80% of CIPK proteins have the function of binding proteins, while the rest proteins have the activity of protein kinase and serine/threonine kinase. In the biological process, CIPK mainly participates in the response to salt stress and dehydration. In addition, CIPK protein is also reported to be involved in auxin transport, high salt response, tissue and organ development, ABA response, cytokinin signal response, and the extended seed coat phenotypes, for example.

### 2.4. Tissue Expression Analysis of Tomato CIPK Gene

We use Genevestigator tomato gene chip platform to analyze data of tomato CIPK gene family expression changes (Figure 4). However, in the gene chip platform, we only found information about 9 tomato *CIPK* genes. The results showed that the expression of *SlCIPK9* was the most significant in flowers and there is some expression in the carpels, fruits, and roots. *SlCIPK1* is expressed in all tissues and organs except the roots. *SlCIPK5* was expressed only in roots, lateral roots, and seedlings. *SlCIPK2* and *SlCIPK19* are expressed in various tissues and organs. *SlCIPK10* and *SlCIPK15* are expressed in other tissues and organs except in leaves.

### 2.5. Expression Analysis of Tomato CIPK Gene under Abiotic Stress

In order to study the response of tomato CIPK gene family to abiotic stress (low temperature, drought, high salt), we treated tomato seedlings with low temperature (4 °C), drought and high salt conditions. qRT-PCR detected the expression of each member of the tomato CIPK gene family. The results in Figure 5 reveal that the expression of *SlCIPK1, SlCIPK4, SlCIPK7, SlCIPK8, SlCIPK10, SlCIPK11*, *SlCIPK16, SlCIPK18, SlCIPK21,* and *SlCIPK22* were significantly up-regulated under cold stress. The expression of *SlCIPK1* increased the most after cold induction, and the expression of *SlCIPK1* after cold induction for 0.5 h was 27.57 times that of untreated leaves at 0 h. The highest expression was found at 3 h, which was 50.91 times of untreated leaves at 0 h. Followed by *SlCIPK18* after 0.5 h at 4 °C stress expression quantity increased 10.04 times. After 3 h, the expression increased by 24.59 times, and then decreased gradually. After 24 h of cold induction, the expression was lower than the control. In the early stages, we conducted data analysis on the low-temperature transcriptome of tomato *CIPK* gene family and the results indicate that the expression of *SlCIPK1, SlCIPK2, SlCIPK15, SlCIPK14, SlCIPK21, SlCIPK3, SlCIPK9, SlCIPK10, SlCIPK7, SlCIPK18, SlCIPK12,* and *SlCIPK22* increased after 12 hours of cold induction; The expression levels of *SlCIPK13, SlCIPK20, SlCIPK11, SlCIPK7, SlCIPK19, SlCIPK4,* and *SlCIPK4* were almost unchanged after cold induction. The expressions of *SlCIPK5, SlCIPK6, SlCIPK8,* and *SlCIPK16* were low in tomato. Therefore, the changes in *CIPK* expression detected at low temperature are consistent with the transcriptome data.

The expression of *SlCIPK6, SlCIPK7* and *SlCIPK8* increased significantly under drought stress. Although the expression of *SlCIPK6* did not change significantly at the initial stage of drought stress, the expression of *SlCIPK6* was 14.74 times higher than that of the control after 12 h induction. *SlCIPK7* and *SlCIPK6* have similar expression patterns under drought induction, and their expression levels are similar to those of the control group at the initial stage of drought stress. After a 12 h induction, the expression was increased by 8.49 times. The results showed that compared with the control (0 h), the differential expression of *SlCIPK8* was apparent after 3 h of drought stress, which is 15.63 times as much as the control. After 9 h, 12 h of drought induction, *SlCIPK8* expression quantity increase slightly. 

The differential expressions of *SlCIPK8* and *SlCIPK9* were evident under high salt stress. Although the expression of *SlCIPK8* did not change significantly in the early stage of high salt stress, after 24 h of high salt stress, its expression reached the highest level, which was 28 times higher than that of control, but its expression was rapidly downregulated after 36 hours of high salt treatment. On the other hand, the change of *SlCIPK9* expressions was also not apparent at the initial stage of high salt stress, but after 36 hours the expression reaches a peak.

### 2.6. Functional Analysis of Tomato SlCIPK1 and SlCIPK8 Silencing in Tomato Abiotic Stress

After qRT-PCR analysis of tomatoes *CIPK* gene family members under three kinds of abiotic stress in the expression of quantity changes, we found that the expression of *SlCIPK1* increases were evident in the early stages of cold stress which mean that SlCIPK1 may play a role in the calcium ion signal transduction pathway of tomato under cold stress. The changes of *SlCIPK8* expression were obvious under cold, drought, and high salt stress, suggesting that SlCIPK8 is involved in three signal transduction pathways including low temperature, drought and high salt in tomato. For this reason, SlCIPK1 and SlCIPK8 were considered valuable for further investigation with the use of virus-induced gene silencing methods to further verify their gene functions.

First, we successfully obtained virus-induced gene silencing (VIGS) vectors of *pTRV2-SlCIPK1*, *pTRV2-SlCIPK8,* and then introduced them into agrobacterium. Ptrv2-pds, the silencing vector of *Phytoene Dehydrogenase (PDS)*, was introduced as the indicator gene of VIGS system, and tomato seedlings were infected by using vacuum infiltration for a span of 10 days. Tomato infected with agrobacterium, a recombinant carrier of ptrv2-pds, began to fade from the veins of the new leaf. The veins begin to whiten from the edge indicating that the virus injected into the plant had formed a systematic infection, SlCIPK1, SlCIPK8 are silenced. Moreover, there was no obvious phenotype change in the negative control plants, indicating that the VIGS system was not toxic to tomatoes (Appendix A). After 17 days of tomato inoculation, real-time fluorescence quantitative method was used to detect the gene expression of tomato seedlings and control after inoculation, actin was used as an internal reference. The results showed that the silencing efficiency of SlCIPK1 was 31.9% and that of SlCIPK8 was 98.2% (Appendix A).

We then treated SlCIPK1, SlCIPK8 silenced plants, and a controlled plant with low temperature (4 °C), drought, high salt stress, and the changes of related physiological indexes were detected after the treatment of 0, 3, 6, 9, and 12 h. We found that Ascorbate Peroxidase (APX) expression in tomato plants increased sharply after 9 h of low-temperature treatment and the control was higher than the gene silencing plants. The results showed that the resistance of the gene silenced plants under cold stress was worse than that of the control plants. The content of Superoxide dismutase (SOD) and Peroxidase (POD) in silencing plants was lower than that of wild-type. After 12 h of cold stress, the content of PRO was significantly lower than that of wild-type (Figure 6).

The APX content of control plants was higher than that of SlCIPK1 silenced plants under drought stress for 12 h, but the APX content of SlCIPK8 silenced plant is the lowest, which showed that the drought-resistant ability of plants with silencing SlCIPK1 and SlCIPK8 genes was lower than that of control plants. In both gene silencing and control, Catalase (CAT) content was first increased and then decreased. At later stages of drought stress, the CAT content of SlCIPK1 and SlCIPK8 gene silenced plants was lower than that of the control. After 3 h of drought stress, the SOD content of SlCIPK1 and SlCIPK8 gene silenced plants was lower than that of the control. Malonic Dialdehyde (MDA) content in silenced SlCIPK8 plants after drought treatment for 12 h was higher than the control. The above data indicated that SlCIPK8 silenced plants were more damaged than the control plants under drought conditions (Figure 7).

At 12 h of high salt stress, the content of APX, CAT, POD, SOD, and PRO in control was lower than that of SlCIPK1/8 silent plants. According to the determination of physiological indexes of silenced plants of SlCIPK1 and SlCIPK8 under high salt stress, the resistance ability of silenced plants of SlCIPK1 and SlCIPK8 in the later stage of salt stress was higher than that of control plants (Figure 8). These results indicate that CIPK1 and CIPK8 play an essential role in tomato’s response to low temperature, drought, and high salt stress. We focused our attention on the function of CIPK1/8 at low temperatures in order to study the low-temperature signal pathway regulated by CIPK1/8. We also made further predictions about its interaction with other proteins through bioinformatics (Appendix A). At the same time, we used yeast two-hybrid to screen proteins that can interact with SlCIPK1/8 at low temperature. The molecular mechanism of CIPK response to low temperature was further analyzed.

## 3. Discussion

Abiotic stress can affect plant growth and restrict the geographical distribution of plants. However, plants have developed a series of physiological and biochemical mechanisms to combat these stresses. CBLs and CIPKs have been reported to form complex signaling networks in plants, which play an important role in response to stress. 

At present, the CIPK gene has been found in many plants: 5 PvCIPK were identified in the bean, 26 Canadian canola BnaCIPKs were found [53], 27 PtCIPK were found in poplar [54], 32 TaCIPK were identified in wheat [35], and 25 SmCIPK were found in eggplant [55]. Also, CIPK gene family was found in algae, moss, ferns, and *Arabidopsis thaliana*. The update of genomic annotations is an important follow-up to sequencing. It is imperative to remove the wrong splicing and pseudogenes and add the newly discovered genes to advance research at the genomic level. Twenty-two SlCIPKs were identified from tomato by bioinformatic methods in this study. In addition to chromosome X, the tomato has further distribution of SlCIPK across the genome, with the largest number of members on chromosome IV. This phenomenon of uneven distribution of CIPK on chromosomes is also reported in millet [9], corn [56], and pepper [57].

CIPK is a group of serine/threonine protein kinases, and the activity of CIPKs is regulated by interactions with CBL calcium sensors [9,12]. To regulate the kinase activity of CIPKs, CBLs often determine the function of CIPKs by recruiting CIPKs to a specific subcellular location, as demonstrated by the action of CIPK24/SOS2 in plasma membrane or tonoplast when interacting with either CBL4/SOS3 or CBL10. The structural information on the catalytic domain of the kinase and the kinase inhibition mechanism are not fully understood. Whether the unbound FISL is blocking the active site or inhibits the enzyme by an allosteric mechanism is unknown. 

CIPKs play a critical role in plant responses to environmental stresses, but the function that mediates the stress is poorly understood. The goal of this study was to explore CIPKs involved in the stress response. CIPK gene family of *Arabidopsis* was studied. At least 25 CIPK genes are confirmed members of the family. Research has shown that the expression of *AtCIPK8* is down-regulated in *chl1-5* mutant which involved in the primary nitrate response [58]. CIPK9 is inducible by potassium deficiency and is vital for low-potassium tolerance in *Arabidopsis* [59]. The expression of a CIPK from *Pisum sativum* was upregulated by cold induction [60].

This study analyzed the expression of each member of the SlCIPK gene family under drought, salt, and cold stress and found that they were all affected by abiotic stress. The expression of SlCIPK1 at low temperature was significantly higher than that of other members, and this inducement showed a change in the expression of SlCIPK1 after the first increase. Similar to many associated signal pathway components and induction patterns, these results suggest that SlCIPK1 may play a vital role in the resistance of tomato to cold stress. The expression of SlCIPK8 is not only induced by cold stress but also induced by drought and high salt in different periods.

To further identify the function of SlCIPK1 and SlCIPK8 in abiotic stress tolerance, we silenced SlCIPK1 and SlCIPK8 by VIGS. Real-time fluorescence quantitative PCR was used to calculate the silencing efficiency of the target gene, and the silencing efficiency of SlCIPK was 31.9%, and that of SlCIPK8 is 98.2%. The silenced plants and control plants of SlCIPK1 and SlCIPK8 were subjected to low temperature, drought, and high salt stress treatment, respectively, and related physiological indexes were detected.

Abiotic stress can generate excessive ROS production, which are toxic to the cell and result in membrane damage and cell death. Therefore, various physiological indices were measured to evaluate oxidative injury between SlCIPK1, SlCIPK8 silenced plants, and control plants under cold, drought, and high salinity stress. MDA is often used to evaluate ROS-mediated membrane damage [61]. In this work, lower levels of MDA suggested less oxidative damage in the control plants under cold stress and drought stress. After 9 h of low-temperature treatment, APX expression in tomato plants increased sharply, and the control was higher than that of the gene silenced plants, indicating that the resistance of the gene silenced plants under cold stress was worse than that of the control plants. Further, POD content in control plants increased significantly and reached the maximum value at 12 h after cold stress treatment. The content of peroxidase in the silenced SlCIPK1 and SlCIPK8 plants was always significantly lower than the control. At 12 h of cold stress, the content of PRO in the control plants increased significantly, which was significantly higher than that of the plants with gene silencing. The APX content of the control plants was higher than that of SlCIPK1 silenced plants at 12 h of drought stress, while that of SlCIPK8 silenced plants was the lowest, indicating that the drought-resistant capacity of the plants with silencing SlCIPK1 and SlCIPK8 was lower than that of the control plants. Increased CAT activity reduces stress-induced ROS damage and protects cells from oxidative stress [62]. At the late stage of drought stress, the CAT content of SlCIPK1 and SlCIPK8 silenced plants was lower than the control. 

Furthermore, the content of POD and SOD in silenced plants was lower than that of control plants. However, the contents of APX, CAT, POD, SOD, and PRO in the control plants were lower than those of SlCIPK1/8 silenced plants under 12 h of high salt stress. According to the determination of physiological indexes of silenced plants of SlCIPK1 and SlCIPK8 under high salt stress, the resistance ability of silenced plants of SlCIP1 and SlCIPK8 in the later stage of salt stress was higher than that of control plants. The above results indicate that SlCIPK1 and SlCIPK8 play a critical role in mitigating the stresses associated with low temperature, drought, and high salt. 

How CIPK1/8 identifies specific changes in Ca^2+^ concentration and how the CIPK interact with multiple CBLs and activate the corresponding downstream response proteins will be further understood. Additionally, understanding the function and regulation relationship between CBL-CIPK signal system and other signal systems in plants, such as Type 2C Protein Phosphatases (PP2C), Calcium-dependent Protein Kinase (CDPK), and Mitogen-Activated Protein Kinase (MAPK) is of significant interest to the research community. The enhanced application of this regulatory network in cultivated crops would be valuable to improve resistance to various kinds of stress contributing to increases in cropping system yields and subsequently, to farmers’ income.

## 4. Materials and Methods

### 4.1. Plant Materials and Treatments

*S. lycopersicum* ‘glamor’ was supplied by Tomato Genetics Research Center (University of California, Davis, USA) and conserved in our lab. Seeds of uniform size were surface-sterilized by soaking in 70% (*v*/*v*) ethanol for 2 mins, then in 2.63% (*w*/*v*) NaCl for 30 mins, and finally rinsed five times with sterile distilled water. *S. lycopersicum* ‘glamor’ was grown at 25 °C, 150 μmol 1/2 Murashige and Skoog (MS) medium (Phyto Tech, La Ciotat, France) (m^−2^ s^−1^), with 16 h light and 8 h dark cycles in growth room for 8 weeks before harvesting. The humidity of the growth house is 60%. Only the top three of youngest terminal leaflet materials were collected from treatment and control. At each time point, leaves were sampled from six plants and pooled together. At different time points, six plants leaves from different groups were taken. The harvested leaves were immediately quick-frozen in liquid nitrogen and kept at −80 °C prior to RNA extraction. For cold treatment, plants were transferred to low temperature (4 °C) with 16 h light and 8 h dark cycles and sampled at 0, 0.5, 3, 6, 9, 12, and 24 h time periods. For salt stress, the roots of plants were soaked with 200 mM NaCl (m^−2^ s^−1^) and sampled at 0, 0.5, 3, 6, 9, 12, and 24 h time periods. For drought stress, the roots of plants were soaked with 20% PEG6000 (1/4 Hoagland nutrient solution) and sampled at 0, 0.5, 3, 6, 9, and 12 h time periods. Samples were collected from the second fully expanded functional leaf on the top of the plant morphology and there were three biological replicates per treatment. We measure the content of MDA according to the MDA test box.

### 4.2. Identification of the CIPK Gene Family in Tomato

The whole nucleotide sequence of tomato was downloaded from the tomato genome database (https://solgenomics.net). tBLASTn analyses with all the *Arabidopsis* CIPKs as queries were adopted to identify the CIPKs in tomato genome database. Stringent criteria (E < 10^−10^, identity > 50%) were used to ensure the reliability of the amino acid sequences. In addition to the sequence conservation, the exhibition of the conserved Pkinase domain and NAF domain was the decisive criterion to include a kinase candidate in the CIPK gene family, to confirm genes accurately encoding CIPK proteins, deduced protein sequences were analyzed for the presence of kinase domains and NAF domain using the program PROSITE (http://www.expasy.ch/prosite/).

### 4.3. Bioinformatics Analysis of CIPK Gene Family

The tomato genome information file was downloaded from the Phytozome database, the chromosome position information of CIPK gene was selected by using the Perl program, and the map of chromosome location was made by using the MapDraw tool. This experiment was undertaken through the MEME (http://meme.nbcr.net/meme/) for online analysis of CIPK conservative domain structure and determination of its conservative motif.

In order to standardize analysis of tomato CIPK gene product, analysis implemented with AgBase v2.00 (http://agbase.msstate.edu/index.html) to perform GO searches of the tomato CIPK were used. The CIPK protein in tomato was analyzed using Blastp and UniProt, selected in the database by GOanna. The results were then converted to genetic correlation files using the GOanna2ga program. Finally, using the GOslim Viewer a protein functional annotation profile was generated.

Using the Genevestigator tomato gene chip platform, the tomato CIPK gene family microarray data was analyzed. In the cluster diagram, the depth of red indicates the strength of gene expression, while white indicates no signal. Hence, the darker the color is, the stronger the signal is.

### 4.4. Quantitative Reverse Transcription Polymerase Chain Reaction (qRT-PCR) Analysis

Total RNA was extracted from plants grown at 25 °C for 3 weeks with TRIzol reagent (Invitrogen, Waltham, MA, USA), and qRT-PCR was performed using the SYBR Green PCR Master Mix kit (TaKaRa, Tokyo, Japan), as previously described [63]. Before the qRT-PCR analysis, 1 μL cDNA was diluted with 9 μL of nuclease-free water. Each PCR reaction was conducted in a 20-μL reaction volume containing 10 μL 2 × qPCR SuperMix (Trans, Beijing, China), 0.4 μL 10 μM solution per primer, 1 μL diluted cDNA and 8.2 μL ddH_2_O. The PCR program was set as follow: 94 °C for 30 s and 40 cycles of 94 °C for 5 s and 60 °C for 30 s. Melting curves was also checked. We compare gene expression between control and treatment done at the same time. Each sample was repeated biologically three times, and the internal reference gene was the housekeeping gene *Actin*. Data represent means of three replicates ± standard deviation (SD). Error bars represent the SD. Asterisks denote a significant difference from the wild-type plants (*, *p* < 0.05; **, *p* < 0.01, Student’s *t*-test) [64]. The primers used are listed in Appendix A. 

### 4.5. Virus-Induced Gene Silencing (VIGS)

Virus induced gene silencing (VIGS) refers to the ability of a plant to induce silencing of endogenous genes and cause phenotypic changes after a virus carrying a target gene fragment infects a plant, and then studies the function of the target gene based on phenotypic variation. Tomato seedlings were removed from the soil and the roots were gently washed. PTRV1, pTRV2, ptrv2-pds, ptrv2-slcipk1, ptrv2-slcipk8 positive agrobacterium gv3pk8 was inoculated in 10 mL Luria Bertani (LB) liquid medium containing Rif (100 g/mL), Kan(50 g/mL), gen(50 g/mL) resistance, and gen(50 g/mL) resistance, respectively, in 28 °C, 200 RPM/min constant temperature incubator in shaking culture 12–16 h. 200 leucine solution was inoculated into 50 mL LB liquid medium containing 2-n-morpholine sulfonic acid (MES) containing 10 mM, 20 M acetylacetone (AS), Rif (100 leucine g/mL), Kan (50 leucine g/mL), Gen (50 g/mL) 28 °C, the constant temperature incubator of 200 r/min shaking culture for the night. The shaken liquid was centrifuged at room temperature for 10 min at 5000 RPM, and the supernatant was discarded. The collected bacterial weight was suspended in LB medium containing 100 mM MgCl_2_, 10 mM MES, and 200 mM AS, respectively, and OD600 was adjusted to about 1.0 (compared with LB without bacteria) and left at room temperature for 3 h. Agrobacterium containing pTRV1 was mixed with agrobacterium containing ptrv2-slcipk1, ptrv2-slcipk8, ptrv2-pds, and pTRV2 respectively at a volume ratio of 1:1 for inoculation. To ensure successful infiltration tomato plant materials were inoculated with different combinations of fluid in each 4 min vacuum process. Afterward, the plants were placed in the incubator to continue to grow at 21/21 °C (light/dark), the temperature was the same for both light and dark regimes. 

## Figures and Tables

**Figure 1 ijms-21-00110-f001:**
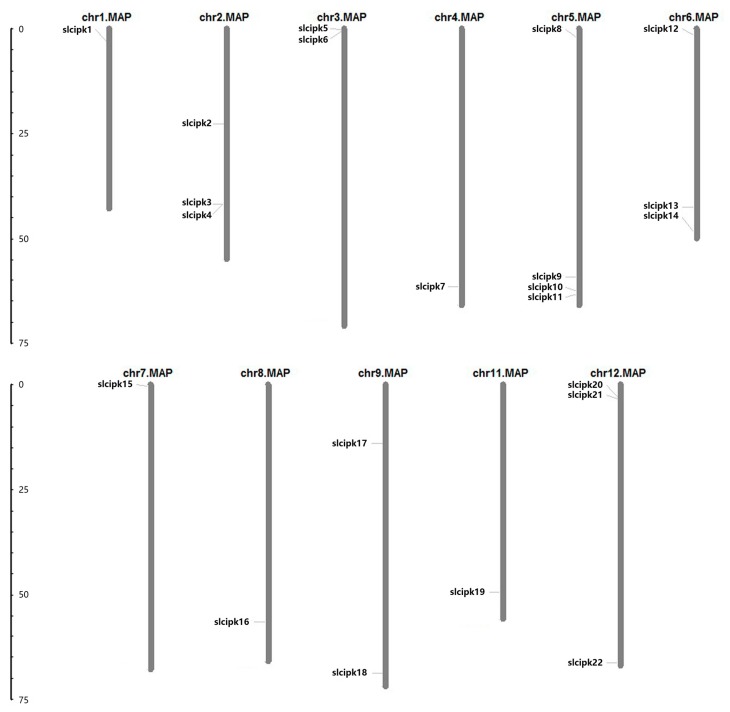
Chromosome mapping of CIPK gene family.

**Figure 2 ijms-21-00110-f002:**
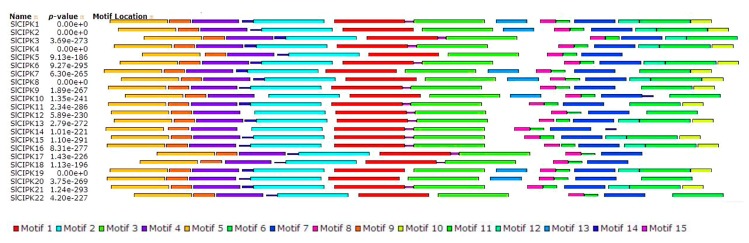
Conserved motifs of tomato CIPK protein. The protein sequences of 22 genes contain 15 conserved motifs, and the same conserved motifs represent the same color.

**Figure 3 ijms-21-00110-f003:**
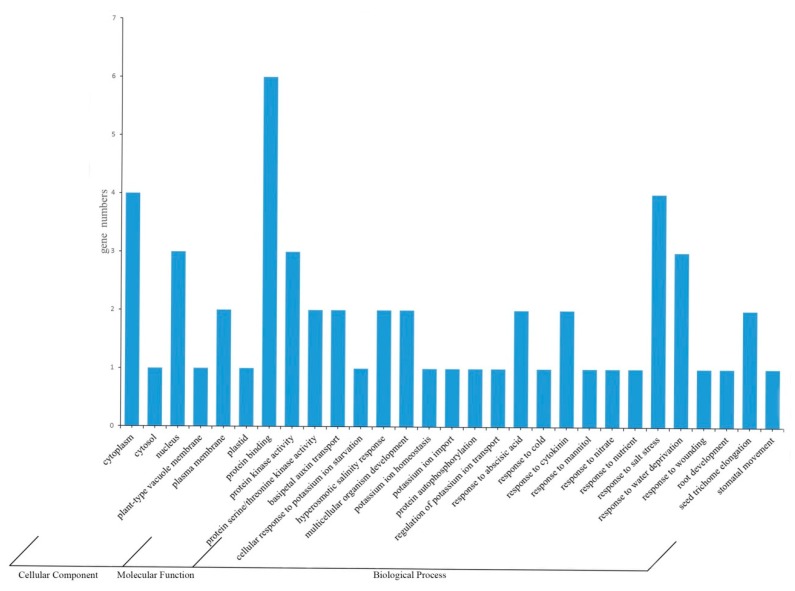
Gene Ontology (GO) analysis output of tomato *CIPK* gene.

**Figure 4 ijms-21-00110-f004:**
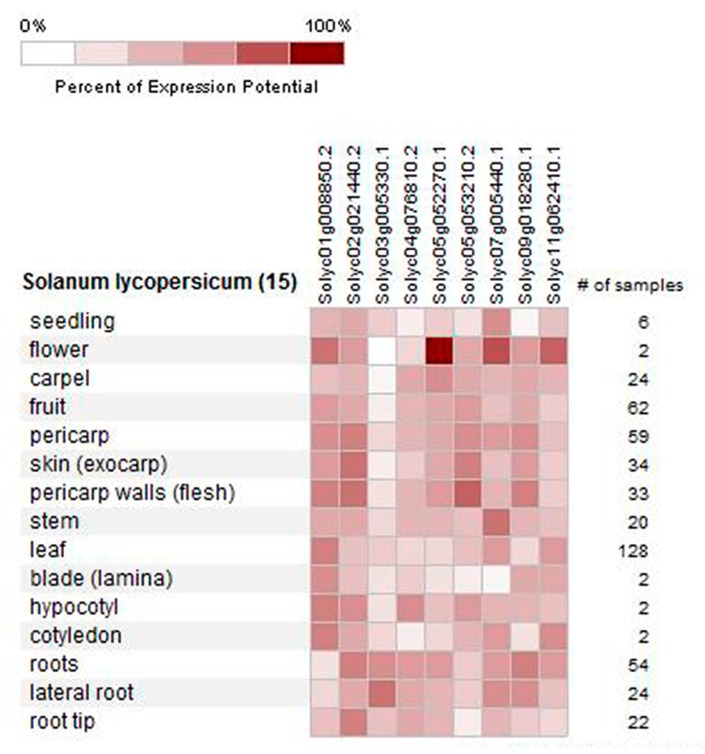
The expression pattern of the tomato *CIPK* gene.

**Figure 5 ijms-21-00110-f005:**
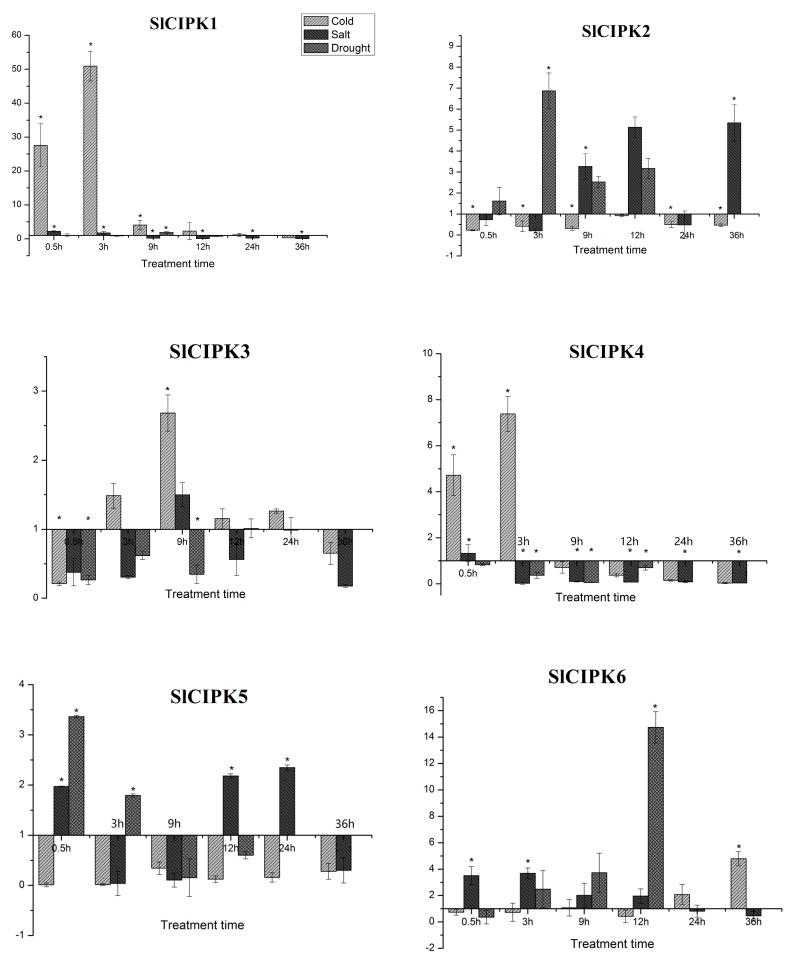
Expression analyses of tomato *CIPK* genes under abiotic stress. Tomato seedlings were treated with low temperature (4 °C), drought and high salt conditions in 36 h. The expression of each member of the tomato CIPK gene family was detected by qRT-PCR (*, *p* < 0.05, Student’s *t*-test).

**Figure 6 ijms-21-00110-f006:**
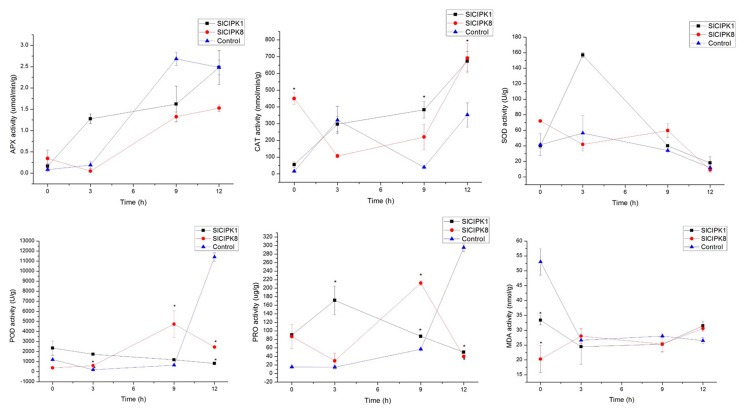
Effects of cold stress on the activities of Ascorbate Peroxidase (APX), Catalase (CAT), Superoxide dismutase (SOD), Peroxidase (POD), Proenzyme (PRO), and Malonic Dialdehyde (MDA) of *SlCIPK1*, *SlCIPK8* silencing tomato, and control. (*, *p* < 0.05, Students *t*-test).

**Figure 7 ijms-21-00110-f007:**
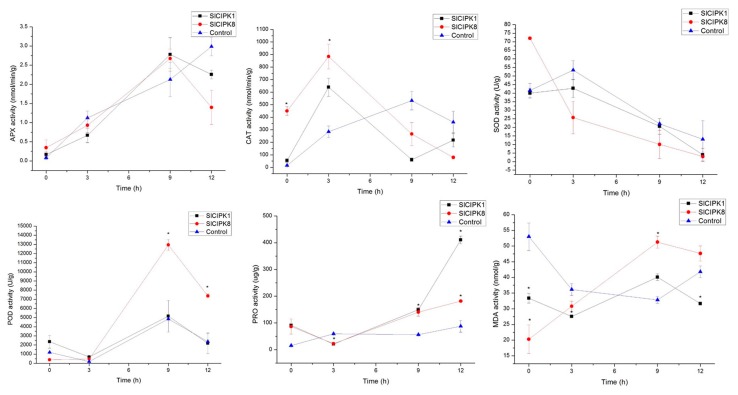
Effects of drought stress on the activities of APX, CAT, SOD, POD, PRO, and MDA of *SlCIPK1*, *SlCIPK8* silenced tomato, and control. (*, *p* < 0.05, Students *t*-test).

**Figure 8 ijms-21-00110-f008:**
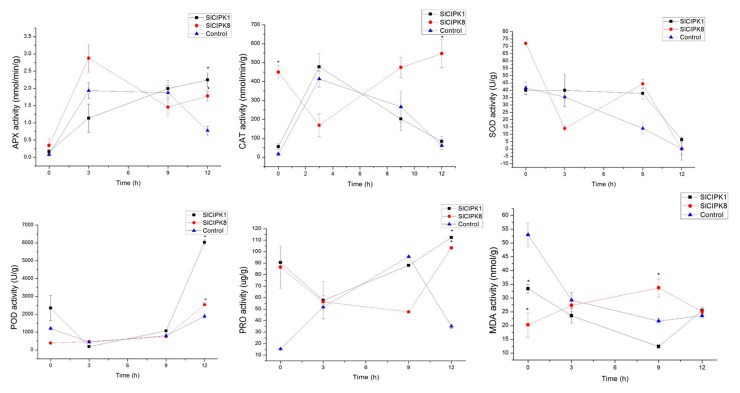
Effects of salt stress on the activities of APX, CAT, SOD, POD, PRO, and MDA of *SlCIPK1*, *SlCIPK8* silenced tomato, and control. (*, *p* < 0.05, Students *t*-test).

**Table 1 ijms-21-00110-t001:** Characteristics of tomato calcineurin B-like interacting protein kinase (CIPK) genes.

Gene Name	Sol Locus	No. Amino Acid	Intron	pI	Protein M.W. (kDa)	TAIR Locus	Gene Name	Percentage Homology (%)
*SlCIPK1*	*solyc01g008850.2.1*	438	14	6.88	50.1	*At2g26980.4*	*AtCIPK3*	82
*SlCIPK2*	*solyc02g021440.2.1*	457	14	8.93	51.3	*At1g30270.1*	*AtCIPK23*	84
*SlCIPK3*	*solyc02g072530.1.1*	478	0	8.81	53.8	*At4g18700.1*	*AtCIPK12*	78
*SlCIPK4*	*solyc02g072540.2.1*	460	1	9.16	52.5	*At5g45820.1*	*AtCIPK20*	67
*SlCIPK5*	*solyc03g005330.1.1*	428	0	9.21	48	*At3g23000.1*	*AtCIPK7*	53
*SlCIPK6*	*solyc03g006110.2.1*	465	1	9.03	57.8	*At5g45820.1*	*AtCIPK20*	66
*SlCIPK7*	*solyc04g076810.2.1*	447	13	6.38	50.6	*At4g24400.1*	*AtCIPK8*	80
*SlCIPK8*	*solyc05g047600.2.1*	451	14	7.69	51.3	*At1g01140.1*	*AtCIPK9*	74
*SlCIPK9*	*solyc05g052270.1.1*	440	0	8.99	49.8	*At5g58380.1*	*AtCIPK10*	63
*SlCIPK10*	*solyc05g053210.2.1*	460	11	6.36	51.9	*At3g17510.1*	*AtCIPK1*	65
*SlCIPK11*	*solyc05g007430.1.1*	478	0	8.69	54.6	*At5g58380.1*	*AtCIPK10*	67
*SlCIPK12*	*solyc06g007440.2.1*	445	1	8.02	50.8	*At2g26980.1*	*AtCIPK11*	59
*SlCIPK13*	*solyc06g068450.1.1*	439	0	9.20	49.7	*At5g25110.1*	*AtCIPK25*	69
*SlCIPK14*	*solyc06g082440.1.1*	419	0	8.71	48.2	*At2g26980.4*	*AtCIPK25*	59
*SlCIPK15*	*solyc07g005440.1.1*	424	0	9.15	47.6	*At4g30960.1*	*AtCIPK6*	74
*SlCIPK16*	*solyc08g067310.1.1*	453	0	8.85	51.8	*At5g10930.1*	*AtCIPK5*	70
*SlCIPK17*	*solyc09g018280.1.1*	443	0	8.36	50	*At5g01820.1*	*AtCIPK14*	62
*SlCIPK18*	*solyc09g083100.1.1*	434	0	8.73	49	*At3g23000.1*	*AtCIPK7*	56
*SlCIPK19*	*solyc11g062410.1.1*	437	0	6.94	50	*At2g26980.4*	*AtCIPK3*	83
*SlCIPK20*	*solyc12g009570.1.1*	446	13	8.86	50.5	*At5g35410.1*	*AtCIPK24*	72
*SlCIPK21*	*solyc12g010130.1.1*	432	0	8.97	48.4	*At4g30960.1*	*AtCIPK6*	72
*SlCIPK22*	*solyc12g098910.1.1*	467	11	6.21	52.8	*At3g17510.1*	*AtCIPK1*	67

The percentage homology refers to the homology with *Arabidopsis* gene.

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
