# Peer review of "Identification and Functional Analysis of Tomato CIPK Gene Family"

_ijms, 2019, doi:10.3390/ijms21010110_

Round 1

Reviewer 1 Report

A global analysis of a particular gene family that has been studied in many other species. They identify stresses that activate some of the genes and work further with two families. This study provides further basic information on the functioning of this gene family but does not take us any further in understanding the mechanism of how the genes help plants avoid stress. Additionally there is too much discussion of future work that is not necessary.

Generally the English needs improving.  Suggest you use a professional editing service. Care must be taken with the placing of reference numbers.

Line 22. Especially apparent?

Line 41. Not all animals can avoid or escape stress, and sometimes the stress is too great or widespread and the animals just can’t get away far enough. It would be better to say ‘many animals’.  In addition, some plants avoid stress (e.g. by turning their leaves, having lifestyle changes that avoid stressful periods) and so it would be better to say ‘many plants’

Line 43. Participate in the signalling during plant growth…

Line 48. Is CML calmodulin like protein?

Line 59 to 71. Paragraph unclear.  Needs rewriting

Line 135. Surely Arabidopsis genes were used to identify the tomato genes?

Line 139. Last sentence is unclear

Table 1. The percentage homology refers to what comparison (nucleotide or amino acid sequences?

Line 172.  Why only 15 out of 22?

Line 197 onwards.  You don’t need to repeat the solyc numbers, they are available from the table

Line 207.  What tissue? The whole seedling?

Fig 5. Did you measure stress responses of any control genes? E.g. genes that are known either to respond to stress or not to respond?

Fig b6. Time courses were very short (12 hours). Did you observe changes in phenotype? What about longer time courses. Did you measure any metabolites (e.g. ascorbate)? Other than MDA.

Line 375 onwards. I take it this is planned work?

Line 385 to 410 should be deleted. You have not reported on this work in results

Line 411 onwards. Also not needed

Line 426. What was the growth medium?  I assume the plants were grown in a growth cabinet?

Line 427. During cold treatment, were the lights on?

Line 430 onwards. Unclear when you applied the drought stress

Line 438. Arabidopsis protein sequences were used as queries?

Line 462. Was DNA destroyed?  Did you check melting curves?

Author Response

Thank you for your suggestions. All your suggestions are very important. They are of great guiding significance to my thesis writing and scientific research.

The following summarizes how we responded to reviewer comments.

Referee #1:

Additionally there is too much discussion of future work that is not necessary.

Answer: According to the referee’s suggestion, we have shorten the content of the discussion, more future work was cut down.

Generally the English needs improving.  Suggest you use a professional editing service. Care must be taken with the placing of reference numbers.

Answer: As suggested, we used a professional editing service to improve our English. We have rechecked and corrected the full text reference numbers.

Line 22. Especially apparent?

Answer: Line 23. We changed "especially obvious" to "relatively obvious".

Line 41. Not all animals can avoid or escape stress, and sometimes the stress is too great or widespread and the animals just can’t get away far enough. It would be better to say ‘many animals’.  In addition, some plants avoid stress (e.g. by turning their leaves, having lifestyle changes that avoid stressful periods) and so it would be better to say ‘many plants’

Answer: Line 42. To be more precise, we have changed it to ”more plants ” and “more animals”.

Thank you for your reminder.

Line 43. Participate in the signalling during plant growth…

Answer: Line 45. We have changed the sentence to “Ca2+ broadly participates in the signalling during plant growth and development”.

Line 48. Is CML calmodulin like protein?

Answer: Line 50. We have changed the sentence to “CML (calmodulin like protein)”

Line 59 to 71. Paragraph unclear.  Needs rewriting

Answer: Line 61 to 72. We have rewritten this paragraph, we have summed up its meaning and tried to write sentences that people can understand.

Line 135. Surely Arabidopsis genes were used to identify the tomato genes?

Answer: Line 135. Arabidopsis genes were surely used to identify the tomato genes. CIPK gene family of Arabidopsis have been studied deeply and can be used as a representative to screen tomato genes.

Line 139. Last sentence is unclear

Answer: Line 139. We have changed the last sentence to” These studies provide new ideas for the further exploration of calcium signaling pathways in tomato under abiotic stress.”

Table 1. The percentage homology refers to what comparison (nucleotide or amino acid sequences?

Answer: Table 1. The percentage homology refers to the homology with Arabidopsis gene. Thank you for your reminding, we have marked this in the original.

Line 172.  Why only 15 out of 22?

Answer: Line 176. The protein sequences of 22 genes contain 15 conserved motifs, and the same conserved motifs represent the same color. Thank you for your reminding again, we have marked this in the original.

Line 197 onwards.  You don’t need to repeat the solyc numbers, they are available from the table

Answer: Line 197 onwards. Thank you for reminding us that we have deleted the duplicate solyc numbers.

Line 207.  What tissue? The whole seedling?

Answer: Line 205. The material we selected was the leaves of tomato seedlings. Thank you for your reminding. We have described it in detail in the materials and methods

Fig 5. Did you measure stress responses of any control genes? E.g. genes that are known either to respond to stress or not to respond?

Answer: Fig 5. We did not measure stress responses of any control genes.

Fig b6. Time courses were very short (12 hours). Did you observe changes in phenotype? What about longer time courses. Did you measure any metabolites (e.g. ascorbate)? Other than MDA.

Answer: Fig 7. We did not observe significant phenotypic changes within 12 hours.

It can be observed from the results that the metabolites detected within 12 hours have relatively obvious changes. In addition, we did not detect other metabolites.

Line 375 onwards. I take it this is planned work?

Line 385 to 410 should be deleted. You have not reported on this work in results

Line 411 onwards. Also not needed

Answer: Line 375 onwards. We have greatly revised the discussion part and deleted the planned work.

Line 426. What was the growth medium?  I assume the plants were grown in a growth cabinet?

Answer: Line 393. S. lycopersicum ‘glamor’ was grown at 25°C, 150μmol 1/2 Murashige and Skoog (MS) medium (m–2 s–1), with 16h light and 8h dark cycles in growth room for 8 weeks before harvesting.

Thank you for reminding us that we have rewritten the materials and methods section and added many details.

Line 427. During cold treatment, were the lights on?

Answer: Line 403. During cold treatment, the lights were on.

Line 430 onwards. Unclear when you applied the drought stress

Answer: Line 405. For drought stress, the roots of plants were soaked with 20%PEG6000 (1/4 Hoagland nutrient solution) and sampled at 0, 0.5, 3, 6, 9, and 12h time periods.

Thank you for reminding us that we have rewritten this section.

Line 438. Arabidopsis protein sequences were used as queries?

Answer: Line 410. Arabidopsis protein sequences were used as queries. Thank you for reminding us that we have added “Arabidopsis” before “CIPKs”

Line 462. Was DNA destroyed?  Did you check melting curves?

Answer: Thank you for reminding us that we have rewritten “Quantitative reverse transcription polymerase chain reaction (qRT-PCR) analysis” and added many details.

Reviewer 2 Report

The manuscript entitled “Identification and functional analysis of tomato CIPK gene family” authored by Yao Zhang , Siyuan Liu , Xi’nan Zhou , Anzhou Yu , Xiuling Chen , Jiayin Liu , Aoxue Wang describes studies of the role of the CIPK family members in abiotic stress responses in tomato. The manuscript requires a careful editing, as well as addition of critical information which is unfortunately missing in the submitted paper.

Below are listed remarks for each of the sections

Abstract

Lane 34-35: “This study has the potential to provide candidate genes in tomato with improved resistance to abiotic stress.” This statement is vague, unclear and should be better reflected in the discussion part. Kindly reformulate it.

Introduction

Through the entire introduction and discussion section, the references are listed in an incorrect way. Please change and enter all references in the manuscript before the punctuation. The reader is confused if the listed reference refers to the sentence before or after punctuation. If it refers to the sentence after punctuation, the sentence shouldn’t be started with a reference, it should be moved towards the end of the sentence. In this manuscript it seems that authors put it after punctuation but referred to the sentence before. Kindly review very carefully and correct. This is a critical point. While correcting refernces, enter space before the sentence and the [number].

Examples of misplaced references are given below:

Lane 42: to adapt to stresses within their growing environment.[1]

Lane 44: environmental stimuli.[2,3]

Lane 47: the plant.[4]

Lane 50: CaMK (calmodulin-dependent protein kinase).[5]

Lane 54: activate downstream targets, thereby regulating the physiological functions of plants.[7-10]

Lane 60: hand structures of high-affinity calcium ions with fixed sequence and spacing.[12-14] Besides

Lane 81-82: high salt stress.[24-26] However, unlike CBL4/SOS3, which functions mainly in the roots[27], 81 CBL10 is expressed and functions almost exclusively in the shoots and leaves[28-29] while

Lane 97: resistance of rice. [39] Sun et al. found that root

Kindly review the entire text and correct English. Ensure that sentences are not too long, as the reader loses the context while reading. Some examples:

Lane 42-43 “Ca2+ broadly participate” change to “Ca2+ broadly participates”

Lane 89: sos3 kindly check if it shouldn’t be SOS3

Lane 90: “Overexpression sCIPK15 can enhance the resistance of 90 rice to salt stress.” Change to “Overexpression of sCIPK15 can enhance the resistance of 90 rice to salt stress.”

Lane 97: “Sun et al. found that root” add year of publication after Sun et al (year). Please review other similar cases in the text.

Lane 115-116: “Reports suggest the role that Arabidopsis CIPK3 CIPK9, CIPK23, CIPK26 have in downstream CBL2/3.” Kindly rewrite this statement.

Lanes 116-120: “Together these proteins regulate the balance of magnesium ions in plants;[52] AtCIPK26 interacts with AtRbohF to inhibit the production of ROS;[53] AtCIPK11 can directly interact with the proton pump on the cell membrane and inhibit the activity of H+-ATP enzyme AHA2, and Atcipk11 deletion mutants increased AHA2 activity and increased plant tolerance to high pH stress;[54] CIPK1 of Arabidopsis interacts with CBL1 and CBL9 respectively to influence ABA expression.[55,56]” Kindly rewrite, shorten sentences, references after “;” refer to the text before the “;” or after? Review again the entire text for similar cases.

Lane 124: “widely planted around in the world” change to “widely planted around the world”

Lane 125-126: “The CBL-CIPK pathway in tomato has made some progress in recent years.” How CBL-CIPK pathway has made some progress in recent years? Did you maybe want to say that research which was conducted in this field has advanced our knowledge about the role the pathway plays in tomato? But in which context, is it in stress responses or other physiological responses of the plant?

Lanes: 139-143: “In order to further explore the protein interacting with SlCIPK1 and SlCIPK8 at low temperature, it provides a theoretical basis for its functional diversity and its recognition, conduction and molecular mechanism of abiotic stress signals, and provides a new idea for further exploring the signaling pathway of calcium ions in tomato at low temperature.” Kindly rewrite! The reader looses the context. Plus kindly edit English.

Results

Lane 144: “2.Results” add space after “2”

Lane 148-149: there is no closing bracket: “(E-value <1e-10,identity> 148 50%, removing duplicates to obtain tomato candidate CIPK.”

Lanes 233-235 “The differential expressions of SlCIPK8 and SlCIPK9 were evident under high salt stress. Although the expression of SlCIPK8 did not change significantly in the early stage of high salt induction, 24h after induction expression quantity to 29.89 times the peak contrast, at 36h express quantity reduce rapidly, and the photography nearly horizontal.” Kindly rewrite. State clearly what is the “high salt induction” as the reader would like to have the information by hand and not look for it in the text. The second part of this text is very confusing: “24h after induction expression quantity to 29.89 times the peak contrast, at 36h express quantity reduce rapidly, and the photography nearly horizontal”. Please phrase the description in such a way that it is clear, and that scientific wording common for such methods is used.

Lane 238: “expressing quantity” verify and use correct description

Lane 261: change (Supplement 1) to (Supplement Figure 1).

Lane 264: change (Supplement 2) to (Supplement Figure 2)

Kindly review the entire text for similar cases as in lane 261 and Lane 264 and correct.

Lanes 279, 280: “In both gene-silencing and control, CAT content was first increased and then decreased. At later stages of drought stress, the CAT content of SlCIPK1 and SlCIPK8 gene silencing plants” Be constant in the way how you use some words through the text e.g. here you use “gene-silencing” as well as “gene silencing”. There are similar cases where two words are connected with “-“ and in other places not. Revise the usage of “gene silencing” term through the entire text. In the context where you describe “gene silencing plants” I would rather apply “gene silenced plants”.

Lane 293: add space after “plants”: “plants(Figure 6c).”

Lane 297: (Supplement figures 3, 4) follow authors guidelines on manuscript preparation and apply consistently through the text, e.g. (Supplement figures 3, 4) or (Supplement Figures 3, 4)

Figures which are presented in this section (Results) as well in the supplementary files are of very bad resolution and quality. Moreover, the figure legend contains the title. If not recommended differently in the author submission guidelines, figure legends must be corrected and must include all necessary information to allow the reader to understand the figure and find all the necessary information without the need of going back to the main text. A figure should be a stand-alone part of manuscript, as said with all necessary information in the legend. This must be corrected as in its current.

Some examples and specific comments are listed below:

Figure 1: This figure is not clear at all. The text is very small and hard to read. What does the star mean which is on some of the chromosomes? Also, the text on the chromosomes (genes) are not visible at all. For better visibility authors could arrange chromosomes in two layers and enlarge them to allow the reader clearly to see the text.

Figure 2: The font difference in this figure. No description of colors, figure not very transparent in its current look.

Figure 4: The font is totally different, both the size and type. It looks as if it was a screenshot or an automated report from a software.

Figure 5: This is the worst of all figures. It is not transparent, chaotic and the message is not easily readable. The legend is a must, especially here. Kindly try to rearrange this figure, or maybe reduce the number of presented results and move some to supplementary files? It would be ideal if the same scale is applied to different graphs, this would require maybe splitting the graphs in various importance groups as some scales go up to 10 others much higher. When rearranging graphs and making new figure, the same size of every graph should be adopted, and care should be taken that bulks and parts of graphs don’t overlap with the text.

Figure 6a and Figure 6b and Figure 6c: Kindly rename as Figure 6, Figure 7, Figure 8 or combine most important results and move some to supplementary files. Labeling figure as Figure 6a or Figure 6b indicates in the text that it is a Figure 6 with different reading panels (a, b, c).

Discussion:

Lane 303: add space after “3”: “3.Discussion”

Lane 306-307: “Among these mechanisms, CIPKs are a crucial family of genes involved in abiotic stress.” Kindly rewrite

Lane 308: “plants: Five PvCIPK” When listed in the text, a number should be given in such a way “plants: 5 PvCIPK”

Lane 314: “22 SlCIPKs” when a number starts a sentence, it should be in a written form: “twenty-two SlCIPKs”

Lane 314: state precisely what you mean by : “by bioinformatics”

Lane 332: add space: “Arabidopsis[70]” review other similar cases.

Lanes 395-410: Kindly shorten this part. While reading, suddenly it appears as if it was a part of results or materials section and not discussion. Therefore, some details regarding experiments should be removed from here. Are these descriptions mentioning future research and future work experiments which were initiated?

Materials and Methods:

This section must be rewritten and information which is usually included in this paragraph should be added. The role of listing material and methods in the manuscript is to avail reader possibility to repeat/do similar experiments (following detailed description) and to get to the same conclusion/result. In this manuscript the information is missing. Some examples (just few):

No description of were plants were grown, in soil, on growth media, in a substrate, in growth room, in an incubator, field? How were plants harvested and why authors mention harvesting of plants prior description of abiotic stresses? How the treatment with NaCl was performed, was it through watering, was it applied once or during the entire period of treatment? How many plants were grown, how many were taken for which treatment and in how many replicates? What are treatment plots mentioned in the description of drought experiment. What were growth conditions during treatment? What was the humidity during all experiments, was FWC measured? How sampling was done? For each timepoint the same plant was used, or the top 3 leaflets were harvested from 6 plants at each timepoint and for the next timepoint the next 6 plants were taken? How many genes were studied, how many primer pairs designed, what was housekeeping genes used in this study, what were qrtPCR conditions. How analyses were performed?

As of Lane 485 no further information as for methods is revealed. What was exactly/and how done with these plants. Which analyses were applied?

Kindly review this section and updated including all relevant information.

Author Response

Thank you for your suggestions. All your suggestions are very important. They are of great guiding significance to my thesis writing and scientific research.

The following summarizes how we responded to reviewer comments.

Referee #2:

Abstract

Lane 34-35. We have changed “This study has the potential to provide candidate genes in tomato with improved resistance to abiotic stress.” to “This study provides a theoretical basis for elucidating the function of tomato CIPK at low temperature and its molecular mechanism of regulating low temperatures.” At the same time, we explain this point in more detail in the discussion.

Introduction

Thank you very much for your suggestion, we have checked the full text and changed the location of all reference serial numbers. Thank you very much for reading this article carefully and making many suggestions. We have changed the following sentences accordingly.

(1) Lane 42-43 “Ca2+ broadly participate” change to “Ca2+ broadly participates”

Line44-45. We have changed “Ca2+ broadly participate” to “Ca2+ broadly participates”

(2) Lane 89: sos3 kindly check if it shouldn’t be SOS3

Line 90.We checked and confirmed “sos3 mutant” is appropriate. Thank you for reading carefully.

(3)Lane 90: “Overexpression sCIPK15 can enhance the resistance of 90 rice to salt stress.” Change to “Overexpression of sCIPK15 can enhance the resistance of 90 rice to salt stress.”

Line91.Thank you for your reminding. We have corrected it.

(4) Lane 97: “Sun et al. found that root” add year of publication after Sun et al (year). Please review other similar cases in the text.

Thank you for your reminding. We have checked the whole article and corrected the cases.

(5) Lane 115-116: “Reports suggest the role that Arabidopsis CIPK3 CIPK9, CIPK23, CIPK26 have in downstream CBL2/3.” Kindly rewrite this statement.

We have changed to “Reports show the roles of Arabidopsis CIPK3 CIPK9, CIPK23 and CIPK26, which regulate the balance of magnesium ions in downstream CBL2/3 plants.”

(6) Lanes 116-120: Kindly rewrite, shorten sentences, references after “;” refer to the text before the “;” or after? Review again the entire text for similar cases.

Line 116-122.We have rewritten and shorten sentences, and checked the full text and changed the location of all reference serial numbers.

(7) Lane 124: “widely planted around in the world” change to “widely planted around the world”

Line 124.Thank you for your reminding. We have corrected it.

(8) Lane 125-126: “The CBL-CIPK pathway in tomato has made some progress in recent years.” 

We have changed it to “CBL-CIPK signaling pathway has been reported in tomato in order to cope with stresses.”

(9) Lanes: 139-143: Kindly rewrite! The reader looses the context. Plus kindly edit English.

Thank you for your suggestion. We have deleted this paragraph and changed it to “These studies provide new ideas for the further exploration of calcium signaling pathways in tomato under abiotic stress.”

Results

Lane 144: “2.Results” add space after “2”

Line 142. Thank you for your reminding. We have corrected it.

Lane 148-149: there is no closing bracket: “(E-value <1e-10,identity> 148 50%, removing duplicates to obtain tomato candidate CIPK.”

Line 146-147. Thank you for your reminding. We have corrected it.

Lanes 233-235. Please phrase the description in such a way that it is clear, and that scientific wording common for such methods is used.

Thank you for your suggestion. We have changed this paragraph to “The differential expressions of SlCIPK8 and SlCIPK9 were evident under high salt stress. Although the expression of SlCIPK8 did not change significantly in the early stage of high salt stress, after 24 hours of high salt stress, its expression reached the highest level, which was 28 times higher than that of control, but its expression was rapidly downregulated after 36 hours of high salt treatment. On the other hand, the change of SlCIPK9 expressions was also not apparent at the initial stage of high salt stress, but after 36 hours the expression reaches a peak.”

Lane 238: “expressing quantity” verify and use correct description

Line 252. Thank you for your suggestion. We have changed it to “expression”.

Lane 261: change (Supplement 1) to (Supplement Figure 1).

Line 268. Thank you for your reminding. We have corrected it.

Lane 264: change (Supplement 2) to (Supplement Figure 2)

Line 272. Thank you for your reminding. We have corrected it. And we have changed the similar cases.

Lanes 279, 280: “In both gene-silencing and control, CAT content was first increased and then decreased. At later stages of drought stress, the CAT content of SlCIPK1 and SlCIPK8 gene silencing plants” Be constant in the way how you use some words through the text e.g. here you use “gene-silencing” as well as “gene silencing”.There are similar cases where two words are connected with “-“ and in other places not. Revise the usage of “gene silencing” term through the entire text. In the context where you describe “gene silencing plants” I would rather apply “gene silenced plants”.

Thank you for your suggestion. We have deleted “-” and changed “gene silencing plants” to“gene silenced plants”.

Lane 293: add space after “plants”: “plants(Figure 6c).”

Line 293. We have added space. Thank you for your reminding.

Lane 297: (Supplement figures 3, 4) follow authors guidelines on manuscript preparation and apply consistently through the text, e.g. (Supplement figures 3, 4) or (Supplement Figures 3, 4)

Thank you for your reminding. We have corrected it.

Figures

Thank you for your suggestion. We have improved the pixel of the picture and added the corresponding legend so that the readers can read it smoothly.

Figure 1. For better visibility we have arrange chromosomes in two layers and enlarge them to allow the reader clearly to see the text.

Figure 2. We have improved the pixel of the picture and added the corresponding legend.

Figure 4. We changed the picture again.

Figure 5. We have revised and rearranged the pictures so that the readers can read them smoothly

Figure 6, 7, 8. We have rename as Figure 6, Figure 7, Figure 8.

Discussion

Lane 303: add space after “3”: “3.Discussion”

Line 311. Thank you for your reminding. We have corrected it.

Lane 306-307: “Among these mechanisms, CIPKs are a crucial family of genes involved in abiotic stress.”Kindly rewrite

Line 314-315. We have changed it to “CBLs and CIPKs have been reported to form complex signaling networks in plants, which play an important role in response to stress.”

Lane 308: “plants: Five PvCIPK”When listed in the text, a number should be given in such a way “plants: 5 PvCIPK”

Line 316. Thank you for your reminding. We have corrected it.

Lane 314: “22 SlCIPKs”when a number starts a sentence, it should be in a written form: “twenty-two SlCIPKs”

Line 322. Thank you for your reminding. We have corrected it.

Lane 314: state precisely what you mean by : “by bioinformatics”

Line 322. We have changed it to “by bioinformatic methods”.

Lane 332: add space: “Arabidopsis[70]” review other similar cases.

Line 340. Thank you for your reminding. We have corrected it.

Lanes 395-410: Kindly shorten this part. While reading, suddenly it appears as if it was a part of results or materials section and not discussion. Therefore, some details regarding experiments should be removed from here. Are these descriptions mentioning future research and future work experiments which were initiated?

Line 375-390. Thank you for your suggestion. We have deleted and shortened this paragraph appropriately.

Materials and Methods

Thank you very much for your suggestion. We have rewritten this part, and here are two paragraphs (we have redescribed the details).

lycopersicum ‘glamor’ was supplied by Tomato Genetics Research Center (University of California, Davis, USA) and conserved in our lab. Seeds of uniform size were surface-sterilized by soaking in 70% (v/v) ethanol for 2 mins, then in 2.63% (w/v) NaCl for 30 mins, and finally rinsed five times with sterile distilled water. S. lycopersicum ‘glamor’ was grown at 25°C, 150μmol 1/2 Murashige and Skoog (MS) medium (m–2 s–1), with 16h light and 8h dark cycles in growth room for 8 weeks before harvesting. The humidity of the growth house is 60%. Only the top three of youngest terminal leaflet materials were collected from treatment and control. At each time point, leaves were sampled from six plants and pooled together. At different time points, six plants leaves from different groups were taken. The harvested leaves were immediately quick-frozen in liquid nitrogen and kept at -80℃ prior to RNA extraction. For cold treatment, plants were transferred to low temperature (4℃) and sampled at 0, 0.5, 3, 6, 9, 12, and 24h time periods. For salt stress, the roots of plants were soaked with 200mM NaCl (m–2 s–1) and sampled at 0, 0.5, 3, 6, 9, 12, and 24h time periods. For drought stress, the roots of plants were soaked with 20%PEG6000 (1/4 Hoagland nutrient solution) and sampled at 0, 0.5, 3, 6, 9, and 12h time periods.

Total RNA was extracted from plants grown at 25°C for 3wk with TRIzol reagent (Invitrogen), and qRT-PCR was performed using the SYBR Green PCR Master Mix kit (TaKaRa, Tokyo, Japan), as previously described [74]. Before the qRT-PCR analysis, 1 μL cDNA was diluted with 9 μL of nuclease-free water. Each PCR reaction was conducted in a 20-μl reaction volume containing 10 μL 2×qPCR SuperMix, 0.4 μL 10 μM solution per primer, 1 μL diluted cDNA and 8.2 μL ddH2O. The PCR program was set as follow: 94 °C for 30s and 40 cycles of 94 °C for 5 s and 60 °C for 30 s. Melting curves was also checked. Each sample was repeated biologically three times, and the internal reference gene was the housekeeping gene Actin. Values are shown as the mean ± standard deviation of three repeats. Student’s t-test was used for statistical analysis [75]. The primers used are listed in Supplementary Table S1.

Round 2

Reviewer 1 Report

The authors have imp[roved the paper but aspects still need improvement

Still minor issues with English and spacing, especially with references

define acronyms (e.g. CIPK and VIGS) at first mention in abstract

Line 33. You did not predict the nucleotide sequence, you uncovered it.

In methods it is not clear how many separate independent biological replicates there are in each treatment. Were the sampled leaves fully expanded or expanding? Were the lights on during cold treatment? How did you measure MDA?  Were comparisons of gene expression between control and treatment done at the same time or was only one time point used for the control. How were statistical comparisons made between control of treatment?

Author Response

Dear Reviewer:

Thank you for the reviewers’ comments concerning our manuscript. Those comments are all valuable and very helpful for revising and improving our paper, as well as the important guiding significance to our researches.

Responds to the reviewer’s comments:

  1. Still minor issues with English and spacing, especially with references

Thank you very much for your reminder. We have revised the references and added spaces where necessary.

  1. define acronyms (e.g. CIPK and VIGS) at first mention in abstract

Thank you very much for your suggestion. We have defined acronyms, for example,” calcineurin B-like interacting protein kinase (CIPK) protein family” and “Virus-induced gene silencing (VIGS)”.

  1. Line 33. You did not predict the nucleotide sequence, you uncovered it.

Line 33. Thank you very much for your reminding. We have changed “predict” to “uncover”.

  1. In methods it is not clear how many separate independent biological replicates there are in each treatment. Were the sampled leaves fully expanded or expanding? Were the lights on during cold treatment? How did you measure MDA?  Were comparisons of gene expression between control and treatment done at the same time or was only one time point used for the control. How were statistical comparisons made between control of treatment?

Thank you very much for your reminder. We have added some details, such as” For cold treatment, plants were transferred to low temperature (4℃) with 16h light and 8h dark cycles”, ” Samples were collected from the second fully expanded functional leaf on the top of the plant morphology and there were three biological replicates per treatment. We measure the content of MDA according to the MDA test box.”, ”We compare gene expression between control and treatment done at the same time. Each sample was repeated biologically three times, and the internal reference gene was the housekeeping gene Actin. Data represent means of three replicates ± standard deviation (SD). Error bars represent the SD. Asterisks denote a significant difference from the wild-type plants (*, P < 0.05; **, P < 0.01, Student’s t-test) [75].”

Reviewer 2 Report

Dear Authors,

Thank you for addressing all points of my review. The manuscript has improved significantly. The materials/methods section could be additionally expanded by detailed description of experiments, which is rather a minor correction. After point 4.5 of materials methods described there is no information on what type of analyses were performed on VIGS plants.

Otherwise, the manuscript must be carefully reviewed for correct English and style (in some places, spaces are missing).

Author Response

Dear Reviewer:

Thank you for the reviewers’ comments concerning our manuscript. Those comments are all valuable and very helpful for revising and improving our paper, as well as the important guiding significance to our researches.

Responds to the reviewer’s comments:

  1. After point 4.5 of materials methods described there is no information on what type of analyses were performed on VIGS plants.

Thank you very much for your reminder. We have added some details, such as” Virus induced gene silencing (VIGS) refers to the ability of a plant to induce silencing of endogenous genes and cause phenotypic changes after a virus carrying a target gene fragment infects a plant, and then studies the function of the target gene based on phenotypic variation.”.

  1. Otherwise, the manuscript must be carefully reviewed for correct English and style (in some places, spaces are missing).

Thank you very much for your reminder. We have revised the manuscript and added spaces where necessary.